



# The KNMI Large Ensemble Time Slice (KNMI-LENTIS)

Laura Muntjewerf[1], Richard Bintanja[1,2], Thomas Reerink[1], and Karin van der Wiel[1]

[1]Royal Netherlands Meteorological Institute (KNMI), De Bilt, The Netherlands
[2]University of Groningen, Energy and Sustainability Research Institute Groningen (ESRIG), Groningen, The Netherlands

**Correspondence:** L. Muntjewerf (laura.muntjewerf@knmi.nl)

**Abstract.** Large-ensemble modelling has become an increasingly popular approach to study the mean climate and the climate system's internal variability in response to external forcing. Here we present the KNMI Large ENsemble TIme Slice (KNMI-LENTIS): a new large ensemble data set produced with the re-turned version of the global climate model EC-Earth3. The ensemble consists of two distinct time slices of 10 year each: a present-day time slice and a +2K warmer future time slice relative to present-day. The initial conditions for the ensemble members are generated with a combination of micro and macro perturbations. The 10-year length of a single time slice is assumed to be too short to show a significant forced climate change signal, and the ensemble size of 1600 years (160×10 years) is assumed to be sufficient to sample the full distribution of climate variability. The time-slice approach makes it possible to study extreme events on sub-annual timescales as well as events that span multiple years such as multi-year droughts and preconditioned compound events. KNMI-LENTIS is therefore uniquely suited to study internal variability and extreme events both at a given climate state and those resulting from forced changes due to external radiative forcing. A unique feature of this ensemble is the high temporal output frequency of the surface water balance and surface energy balance variables, which are stored in 3-hourly intervals, allowing for detailed studies into extreme events. The data set is particularly geared towards research in the land-atmosphere domain. EC-Earth3 has a considerable warm bias in the Southern Ocean and over Antarctica. Hence, users of KNMI-LENTIS are advised to make in-depth comparisons with observational or reanalysis data especially if their studies focus on ocean processes, on locations in the Southern Hemisphere or on teleconnections involving both hemispheres. In this paper, we will give some examples to demonstrate the added value of KNMI-LENTIS for extreme and compound event research and for climate-impact modelling.





# Contents



# 1 Introduction

Climate change is a topic of high societal interest due to its influence on weather (impacts) around the world. Further scientific
understanding of the changing nature of the relationship between weather and society is required to design adequate local
adaptation and mitigation strategies. Not only the climatic mean, but also the variability around the mean is subject to change.
Recent studies have shown that long-term trends in climate variability can differ substantially from trends in mean climate
(Brown et al., 2017; Pendergrass et al., 2017; Bintanja et al., 2020; van der Wiel and Bintanja, 2021), because physical processes
that govern changes in variability can differ from those that affect changes in the mean state. The effects on climate extremes
can even be of the opposite sign (Schaeffer et al., 2005; van der Wiel and Bintanja, 2021), depending on e.g. the climate
variable and the region. Climate models are important tools to examine the Earth system's response to greenhouse gas forcing
and the the associated uncertainties. Model simulations extend and complement the comparatively short observational records.
Further, simulations allow for experiments to test the impacts of specific climate feedback mechanisms, which would not be
possible in the real world.

Single-model initial-condition large ensemble (SMILE) climate model simulations are uniquely suited for the study of
uncertainties in (changing) climate variability and of climate extremes (Deser et al., 2020; Wood et al., 2021). SMILEs consist
of many repetitions of the same climate modelling experiment that only differ in their initial conditions. The different initial
conditions lead to divergence due to the chaotic nature of the climate system, i.e. unpredictable internal variability. This results
in various model realisations within the internal variability of a certain average climate state. The use of SMILEs has become
increasingly popular in climate science, and very recently also started to find its way into other related geosciences, (e.g. in
hydrology, van der Wiel et al., 2019c; Champagne et al., 2020; Poschlod et al., 2020). Typically, large ensemble data sets are set
up following transient climate forcing scenarios, e.g. those designed for the Coupled Model Intercomparison Projects (CMIP).
The choices for emission scenario, simulation length, horizontal and vertical model resolution, and the number of ensemble
members (e.g. Milinski et al., 2020) are often an optimisation between the available computational resources and the need or
wish for more detailed simulations. Various climate modelling centres have produced large ensembles and have made efforts
to make them openly available for research. Examples are the seven CMIP5-class transient ensembles collated in a centralized
archive (MMLEA, Deser et al., 2020), GCM ensemble experiments based on CMIP protocol (e.g. Kay et al., 2015; Maher
et al., 2019; Rodgers et al., 2021; Wyser et al., 2021) and ensembles of regional climate model runs (e.g. Lenderink et al.,
2014; Massey et al., 2015; Leduc et al., 2019; **?**).

In this manuscript we present and describe a recently produced large ensemble following a time slice protocol: the Royal
Netherlands Meteorological Institute (KNMI) Large ENsemble TIme Slice (KNMI-LENTIS). The time slice protocol is dif-
ferent from the transient ensemble data sets mentioned above. We ran many simulations of a decade long for a climate state of
interest rather than a number of multi-decadal or multi-centennial transient simulations. The KNMI-LENTIS ensemble consists
of two time slices: a present-day period and a future period 2 K warmer than present-day. Each time slice has 160 members
of 10 simulations years each. The 10 year segments are assumed to be too short to show a significant forced climate change
signal. Further, we assume that 1600 years of data is sufficient to sample the full distribution of climate variability. Differences



between the two time slices can be attributed to forced climate change. With these assumptions, a single time slice can be used to investigate internal climate variability of a certain climate state, whereas the two time slices together can be applied to study differences in the mean state and the differences in variability between the two climate states.

The KNMI-LENTIS design protocol is inspired by a previous time slice large ensemble produced at KNMI (van der Wiel et al., 2019c) though improved based on earlier experience: longer simulation length (10 years vs. 5 year), higher temporal resolution (sub-daily vs. daily output of surface hydrology and surface energy variables), improved method of micro-perturbations (perturbed initial conditions vs. stochastic perturbed physics tendencies) using the latest release of EC-Earth (CMIP6 generation vs. CMIP5) with higher resolution and improved physics in many aspects (Döscher et al., 2021). The previous large

ensemble has been widely used, for example contributing to analyses of climate variability and forced trends therein (Blackport et al., 2019; van der Wiel and Bintanja, 2021; Sperna Weiland et al., 2021), analyses of changing climate extremes (Bonekamp et al., 2021; Nanditha et al., 2020), and climate attribution research (e.g. Philip et al., 2019, 2020; Kew et al., 2021). Derived data sets, in which the ensemble was used to drive models from other geosciences disciplines, e.g. hydrological modelling (e.g. van der Wiel et al., 2019c; van Kempen et al., 2021), vegetation modelling (e.g. Tschumi et al., 2021, 2022), crop modelling

(e.g. Vogel et al., 2021; Goulart et al., 2021; Zhang et al., 2022) or energy modelling (e.g. van der Wiel et al., 2019a, b), were used to assess the influence of (changing) climate variability on various natural and societal systems. Finally, the data set was used to develop and test scientific methods (e.g. van Kempen et al., 2021; van der Wiel et al., 2021; Boulaguiem et al., 2022).

The way the ensemble is set-up and generated is described in Section 2. In Section 3 we provide a description of the data and discuss the advantages and limitations of the underlying assumptions. In addition, we show examples of possible analyses

using time slice large ensembles, including their value for compound event research as well as for climate-impact modelling (Section 4). Finally a short conclusion is provided (Section 5).

## 2    Set-up

KNMI-LENTIS consists of two time-slices with each 160 simulations of 10-year length. The time-slices represent the present-day climate (2000-2009) and a +2 K warmer future climate (2075-2084 in SSP2-4.5 in EC-Earth3) (Figure 1a). Each time

slice thus consists of 1600 years of model data. In this section we describe the climate model, we elaborate on the choice of the periods and their forcing scenario, and we describe the initial conditions of the individual simulations and how they have been generated. All simulations have a unique ensemble member label that reflects the forcing, the parent and the seed. Further, all simulations are labeled per the CMIP6 CMOR convention of variant labelling. In Appendix A, both the ensemble member label and the CMIP6 variant label of KNMI-LENTIS simulations are explained. The initial conditions (ICs) of the

ensemble members can be characterized by two aspects: the parent simulation from which each member is branched off (macro perturbation), and the seed number which relates to a particular micro-perturbation in the initial three-dimensional atmosphere temperature field.



## 2.1 Model description

KNMI-LENTIS is generated with the model EC-Earth3. EC-Earth3 is a fully coupled, state-of-the-art global Global Climate
Model that is maintained by a consortium of European weather and climate centers (Döscher et al., 2021). The model runs at a
80 km nominal resolution and has prognostic component models for atmosphere, ocean, sea ice, and land hydrology processes.
The atmosphere is simulated with ECMWF's IFS cy36r4. The horizontal resolution is the TL255 spherical harmonics field,
which is linearly reduced in the post processing stage to a Gaussian grid equivalent of 512 x 256 grid cells in longitude/latitude.
The vertical resolution is 91 levels with the top level at 0.01 hPa. The ocean model NEMO3.6 uses a tripolar grid ORCA1 which
primarily has 1° horizontal resolution with meridional refinement down to 1/3° in the tropics. The grid dimensions are 362
x 292 longitude/latitude in the horizontal and 75 levels in the vertical with the top grid cell in the 0-1 m layer. The sea-ice
model is LIM3, which shares the ORCA1 grid. The internal time step for both atmosphere and ocean is 45 minutes. The
coupling frequency between atmosphere and ocean is equal to the internal time step. Further details on the EC-Earth3 model
are provided in Döscher et al. (2021).

The land surface model of EC-Earth is H-TESSEL: Hydrology - Tiled ECMWF Scheme for Surface Exchanges over Land,
with revised land surface Hydrology (van den Hurk et al., 2000; Balsamo et al., 2009; Dutra et al., 2010). H-TESSEL computes
the land surface water and energy balance at the interface of the soil and the atmospheric boundary layer. The model uses a
tiling approach to calculate the surface energy fluxes, the skin temperature and soil parameters. It divides each grid box into
homogeneous fractions (tiles) representative of vegetated, bare soil, frozen water, and liquid water surfaces. The grid box
fluxes and skin temperature values are generated as weighted averages of the tiles. Soil properties and parameterizations are
not tile-specific but instead they apply to the entire grid cell, such that H-TESSEL simulates soil moisture per grid cell.

The EC-Earth3 version that is used for the simulations of KNMI-LENTIS is the *knmi23-dutch-climate-scenarios* project
branch (physics index p5), from now on referred to as the 'ECE3p5 version'. The ECE3p5 version is a re-tuned version of the
EC-Earth 3.3. release for CMIP6 (Döscher et al., 2021). EC-Earth 3.3 has a warm bias in the Southern Ocean and a cold bias
in the Northern Hemisphere. The KNMI re-tuning effort focused on reducing the Northern Hemisphere cold bias. This has
been successful with the trade-off of increasing the Southern Ocean warm bias, therefore introducing a positive global mean
surface temperature (GMST) bias (see also Section 3.2). As the main research aims of KNMI-LENTIS are oriented towards
the Europe region, we have accepted this trade-off.

The re-tuning used a subset of the atmospheric cloud tuning parameters that were selected in earlier work of atmospheric
tuning of EC-Earth3 (see section 2.2.1 in Döscher et al., 2021, for further details). Two tuning parameter values have been
changed in the ECE3p5 version compared to the EC-Earth 3.3.3.2 release: RVICE (fall speed of ice particles) and RLCRIT-
SNOW (critical autoconversion threshold for snow in large-scale precipitation). RVICE is set to 0.1328 in the re-tuning; (0.137
in EC-Earth 3.3.; 0.15 in IFS cy36r4) and RLCRITSNOW is $4.6 \times 10^{-5}$ ($4.0 \times 10^{-5}$ in EC-Earth 3.3.; $5.0 \times 10^{-5}$ in IFS
cy36r4). The other tuning parameters remain the same as in Table 6 of Döscher et al. (2021).

Spinning up the model was done in parallel with the re-tuning process. The spin-up runs of the differently tuned models have
been combined, because the slow ocean spin-up is believed to benefit from running more years with a set of parameter values



that is very close. The initialization of the ECE3p5 version pre-industrial (PI) run was done with the restart files of year 2750 from the EC-Earth3 physics index p2 PI run. The ECE3p5 version PI ran from model year 2750 to 4585. The 16 historical simulations are branched with intervals in the initial conditions of 25 years, starting with member 1 in year model 4550, then going backwards with member 2 in 4525, member 3 in 4500, ... member 16 in 4175. This means for the member with shortest spin-up time with the ECE3p5 version, this is $4550 - 2750 = 1800$ years. Additional spin-up has taken place prior, albeit via runs with a slightly different tuning parameter sets p2 and p1 (Reerink et al., in prep). All added together, the trajectory of the ECE3p5 version spin-up covers about 6000 years.

With this version of EC-Earth, the KNMI has produced an ensemble of 16 transient simulations with CMIP6 forcing (historical, SSP1-2.6, SSP2-4.5, SSP3-7.0, SSP5-8.5). Details of these simulations and of the re-tuning procedure of ECE3p5 version are described in Reerink et al. (in prep).

## 2.2  Time-slice choices: period and forcing scenario

Two choices in the ensemble design have been made a priori: the simulation length and the climatic states of interest. The length of each time-slice was chosen at 10 years. Limiting the simulations to 10 years avoids having any appreciable trend. This approach allows for studying extreme events on subannual timescales as well as events that span multiple years (e.g., multi-year droughts, preconditioned compound events as in Pascale et al. (2021); van der Wiel et al. (2022)). The climatic states of interest are the present-day (named 'PD'), and a future world that is +2K warmer than present-day in the annual annual GMST (named '2K').

The present-day climate in KNMI-LENTIS is represented by the years 2000-2009 simulated using historical forcing. This choice was mainly governed by the availability of initial condition files (every 10 year) and the wish to use CMIP6 historical forcing (available for the years 1850–2014). In Figure 1a the PD time slice is marked by the left pink band.

The main factor in deciding the optimal scenario and years for the 2K climate is the decadal climate change trend. For the purpose of this ensemble it is important that the forced signal within a time slice is as small as possible. This way we can accept each individual year as a suitable representative of the respective climatic state. We have taken the years 1985-2014 from the 16 transient historical simulations with the ECE3p5 version to calculate the present-day GMST. The mean present-day GMST of the 16 members $\mu = 15.47°C$, with an ensemble standard deviation of $\sigma = 0.15°C$. Next, we calculate in what year the annual mean GMST reaches PD+2K in the scenario simulations, and how much GMST changes in the next 10 years. We find that the SSP1-2.6 scenario does not reach PD+2K before 2100. Between the SSP2-4.5, SSP3-7.0 and SSP5-8.5 scenario's, the SSP2-4.5 scenario shows the relatively smallest forced signal in the 10-year period after reaching PD+2K. There is variability among the ensemble members for the exact simulation year when PD+2K is reached Because the forced signal is comparatively small, the SSP2-4.5 scenario shows a relatively larger spread in the timing of reaching the warming target ($\sigma = 9.5$ year). We consider the advantage of a small 10-year forced signal to outweigh this downside, and therefore we choose to initialize the 2K time slice from the SSP2-4.5 scenario. The ensemble-mean simulated year when the forced signal leads to a mean state of PD+2K is year 2073 in the SSP2-4.5 scenario. The 2K climate is represented in KNMI-LENTIS by the years 2075-2084 of the



SSP2-4.5 scenario, again chosen according to the availability of restart files. In Figure 1a the 2K time slice is marked by the right pink band.

## 2.3    Initial conditions

There are several ways to generate multiple unique ensemble members. These include applying micro and macro perturbations to the initial-conditions (Deser et al., 2020). A micro perturbation means adding a round-off level perturbation to an input

field of the GCM, which generally is the three-dimensional atmosphere temperature field. The perturbation propagates due to the chaotic behaviour of the atmosphere model. As such this method produces a unique ensemble member for each unique micro perturbation. Macro perturbations refer to initial conditions that are more fundamentally different among each other. Usually such initial conditions are acquired by branching from a different point in the parent run (like the initialization of the historical simulations described in Section 2.1). This way, the initial state is different not only in the atmosphere but in all

model components. Another method to create different members is to make use of uncertainty in parameter space, for example using stochastic perturbed physics tendencies (SPPT, as used in the operational ECMWF forecast ensemble (Ollinaho et al., 2017; Lock et al., 2019) and in the previous KNMI time slice ensemble by van der Wiel et al. (2019c)).

For KNMI-LENTIS we use a combination of micro and macro perturbed initial conditions. We realise this may impact the ensemble variability of for example the first year; this is investigated in Section 3.5. The parents from which the simulations

are branched can be considered as macro perturbed initial conditions, given that all parents are rooted in the same PI spin-up. The parents are sixteen full transient historical and SSP2-4.5 simulations made using the ECE3p5 version. The 16 historical simulations start in 1850, branched off the PI spin-up simulation. The 16 SSP2-4.5 simulations start in 2015, from the end of their respective historical simulation in 2014 (Figure 1a).

Micro perturbations are applied to the initial three-dimensional temperature field of the atmosphere as in Haarsma et al.

(2020). Each value in the input field is multiplied by a number from a random uniform distribution of values between $-5 \times 10^{-5}$ K and $+5 \times 10^{-5}$ K to yield the perturbed field. We have created nine different random distributions, using a seed from 1 to 9 to assure reproducibility. For one parent, that yields nine sets of micro-perturbed ICs. Including the original set this is ten sets of ICs in total, for ten members (visualized in Figure 1b). Figure 1c visualizes the full ensemble set-up with all members: ten micro perturbations for sixteen historical and scenario macro perturbations, each simulation is run for ten years.

# 3    Limitations

In this section, we evaluate and discuss several aspects of the ensemble that are important for users to consider. We quantify the ensemble temperature difference between the present-day time slice and the +2K time slice for different seasons and for several subsections of the world. We discuss the magnitude of the near-surface temperature biases in the model by making a comparison with ERA5 reanalysis data (Hersbach et al., 2020). Further, we discuss the validity of two critical assumptions:

1. Within a time slice, the 10 year segments are too short to show a significant forced climate change signal,





**Figure 1. Overview of KNMI-LENTIS ensemble set-up.** (a) Global Mean Surface Temperature (GMST) of the 16 ECE3p5 ensemble members forced with CMIP6 historical and SSP2-4.5 forcing. Pink shading shows the two time slices in KNMI-LENTIS. (b) Part of the time slice set-up. From each of the 16 parent runs (grey), 10 KNMI-LENTIS simulations (pink) are branched using unique seeds to make a micro-perturbation in the atmospheric initial conditions. (c) The full ensemble consists of two time slices of 10 years with 1600 years of data each: present-day (PD) and present-day +2K global warming (2K). The parents are visualized by the grey (historical) and blue (SSP2-4.5) lines. The KNMI-LENTIS simulations are visualized by the pink lines.





**Table 1. Quantification of temperature difference and ensemble spread.** Ensemble mean and standard deviation in brackets of the near-surface air temperature difference [K] between the 2K time slice and the PD time slice. Top row: global mean, middle row: Northern Hemisphere mean (180°W –180°E; 0–90°N), bottom row: Europe mean (10°W–40°E; 35–70°N).

|  | ANN | DJF | MAM | JJA | SON |
|---|---|---|---|---|---|
| Global | 1.95 (1.35) | 2.02 (1.51) | 1.83 (1.39) | 1.90 (1.23) | 2.06 (1.26) |
| N.Hemisphere | 2.52 (2.52) | 2.67 (1.99) | 2.29 (1.72) | 2.42 (1.37) | 2.72 (1.49) |
| Europe | 2.55 (2.55) | 2.40 (2.58) | 2.29 (2.16) | 2.81 (1.86) | 2.69 (1.90) |

2. The ensemble size, 1600 years for both time slices, is sufficient to sample the full distribution of climate variability.

If true, this means that a single time slice can be used to investigate internal or natural climate variability at a given climatic state, and that any differences between the two time slices can be attributed to forced climate change. Finally, we comment on the micro and macro initialization method and the legacy effect of a common parent on variability.

## 3.1 Ensemble spread of temperature difference between present-day and +2K

The 2K time slice is designed to be +2K warmer than the GMST of the present-day time slice, as detailed in Section 2.2. Here we quantify the ensemble spread of the temperature differences between the two time slices. The annual GMST in the 2K time slice is on average 1.95 K warmer than the PD time slice (Table 1). The standard deviation over ensemble members of the temperature difference $\sigma$=1.35, and this spread is shown in Figures 7a. Further, the 2K time slice shows enhanced warming in the Northern Hemisphere and Europe (+2.52 K and +2.55K, respectively) and wider ensemble spreads than the global mean values (Table 1).

## 3.2 Quantification of model bias

The re-tuned EC-Earth3 ECE3p5 version has a known warm bias in the Southern Hemisphere (Section 2.1). In this section we quantify the magnitude of this bias and evaluate its spatial and temporal properties. The near-surface temperature of the present-day time slice of the ensemble is compared to ERA5 (Hersbach et al., 2020). We have chosen the 30-year reference period 1990-2019, to have sufficient present-day climate variability to compare with. The ERA5 data is re-gridded to the coarser EC-Earth grid using nearest neighbour interpolation.

The global annual mean surface temperature bias in the PD time slice of KNMI-LENTIS with respect to ERA5 is 1.32 K (Table 2). This is largely due to a strong warm bias in the Southern Ocean and over Antarctica (Figure 2a). The temperature bias over land is generally smaller in magnitude and more often insignificant compared to the ocean bias. The near-surface temperatures in the North Atlantic gyre and the North Pacific gyre are significantly underestimated. In Europe we find a cold bias over Scandinavia, and a warm bias in the region north of the Black sea (Figure 2b).



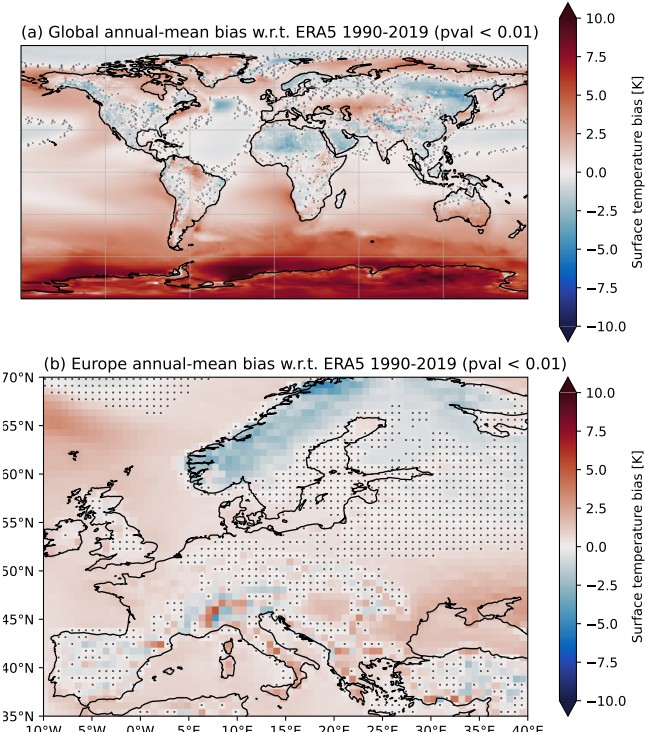

**Figure 2. Model bias.** Near-Surface Air Temperature bias [K] for the annual-average, ensemble-mean of the KNMI-LENTIS present-day time slice [160×(2000-2009)] compared to ERA5 [1990-2019]. Grid cells with a non-significant difference are dotted (p<0.01). For (a) Global and (b) Europe.

The integrated annual mean temperature bias of both the Northern Hemisphere and of Europe are much smaller than the global bias (0.26 K and 0.38, respectively). The warm bias is largest in the June-July-August and in the September-October-November seasons (Table 2).

This outcome is in line with the expectations of using the re-tuned ECE3p5 version, of which the global warm bias is larger than that of the EC-Earth3 released version in Döscher et al. (2021), but much improved for the Northern Hemispere and Europe. Future users of KNMI-LENTIS are advised to make in-depth comparisons with observational or reanalyses data especially if their study focuses on ocean processes, on locations in the Southern Hemisphere or on teleconnections involving both hemispheres.

### 3.3 Forced climate signal within a time slice

To quantify the relative size of the forced climate change signal to the interannual variability within the time slices (assumption 1) we investigate the linear trend of GMST (Figure 3a,c). Both time slices have an interannual ensemble standard deviation of 0.17 K for the annual mean GMST value on the 10 time slice years average. The linear trend over the 10 year simulation



**Table 2. Model bias.** Ensemble mean and standard deviation between brackets of the near-surface air temperature bias [K] of the KNMI-LENTIS present-day time slice w.r.t. ERA5 1990-2019. Top row: global mean, middle row: Northern Hemisphere mean (180°W –180°E; 0–90°N), bottom row: Europe mean (10°W–40°E; 35–70°N). Standard deviation is calculated over the mean biases of all years and all ensemble members, then averaged.

|              | ANN         | DJF          | MAM          | JJA         | SON         |
|--------------|-------------|--------------|--------------|-------------|-------------|
| Global       | 1.32 (0.97) | 1.24 (1.10)  | 1.22 (1.00)  | 1.45 (0.88) | 1.36 (0.93) |
| N.Hemisphere | 0.26 (0.26) | 0.22 (1.46)  | -0.26 (1.24) | 0.44 (0.97) | 0.64 (1.12) |
| Europe       | 0.38 (0.38) | -0.10 (1.99) | -0.06 (1.62) | 0.91 (1.38) | 0.76 (1.44) |

period is only slightly larger than this, at 0.20 K per 10 year for the PD ensemble and 0.22 K per 10 year in the 2K ensemble (black error bars in Figure 3a,c), approximately 1.25 times larger than the interannual ensemble standard deviation. At the global scale, we therefore conclude this assumption holds. Locally, or for other variables, forced trends may exceed the respective trends.

In Figure 3b,d we show the ratio of the 10 yr linear trend of near-surface temperature (TAS) and the ensemble standard
deviation at grid point level. Low values of this quantity are preferred. In most parts of the world, the value is smaller than one, indicating a smaller forced trend than ensemble internal variability. We acknowledge that interannual variability is different for different variables. We therefore advise future users of the KNMI-LENTIS data set to check the validity of assumption 1 on a case-by-case basis.

### 3.4 Range of sampled internal variability

To test whether or not the full distribution of climate variability has been sampled in a 1600 year time slice (assumption 2) we investigate daily temperature variability in a single grid point (52.3°N, 4.9°E; nearest point to De Bilt, the Netherlands). We cannot test how different the distribution would be in a second set of 1600 years. Therefore we investigate differences between two halves of the PD ensemble, each of 800 simulation years. The shape of the distribution in Figure 4a shows two peaks, which is a known phenomenon in The Netherlands due to the fairly rapid transition between the summer and winter
season. The distribution of all daily temperature values in the two smaller ensembles are indistinguishable from each other, suggesting that variability has been adequately sampled in these half ensembles (Figure 4a). Sampling adequately in the tail of the distribution, e.g. for the warmest day of the year, requires larger sampling sizes. However, also here the two halves of the PD ensemble are statistically similar (Fig 4b, differences between the generalized extreme value (GEV) fitted distributions are well within the associated error bars). Comparing the PD distributions to the distribution of the 2K ensemble, we note that
forced climate change does significantly impact the shape of the distribution. We also note that variability beyond the scope of the climate model (e.g. at scales smaller or larger than resolved, missing processes) is not captured by these ensembles.




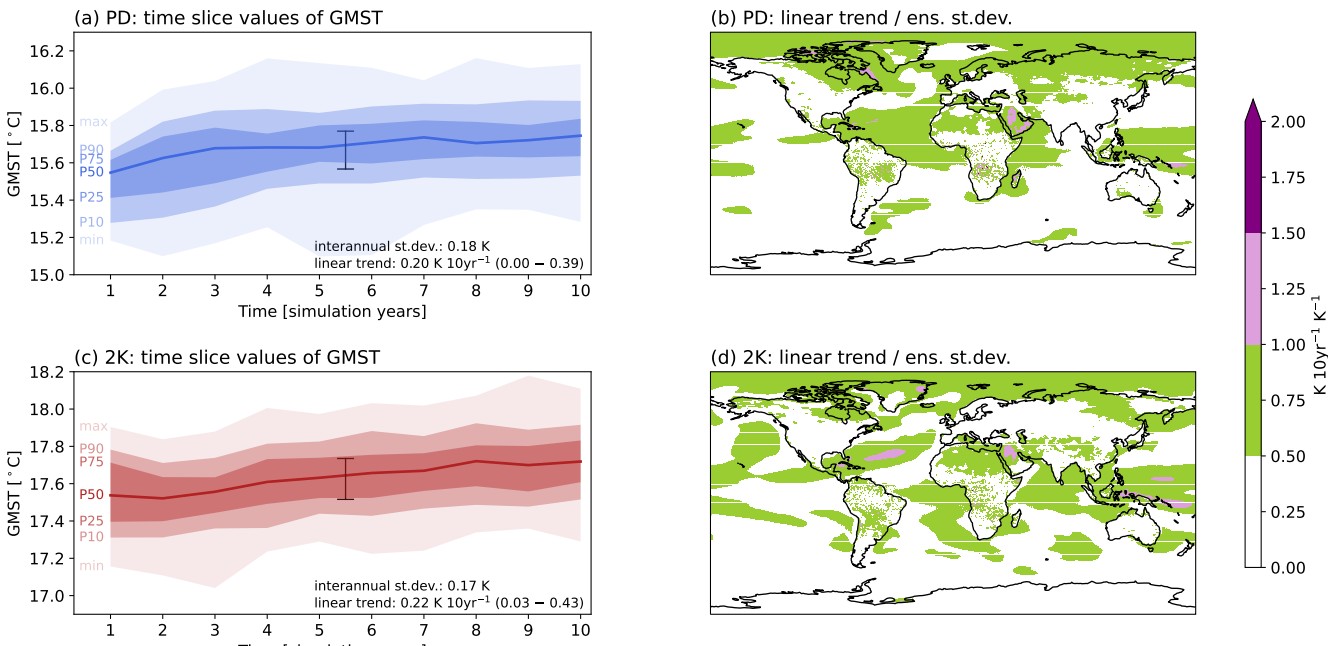

**Figure 3. Quantification of assumption 1: size of forced trends within a time slice.** (a,c) Ensemble spread of annual mean values of Global Mean Surface Temperature (GMST, shaded colours, percentile values denoted on the left). Ensemble interannual standard deviation and ensemble linear trend of GMST over 10 years shown in bottom right corner, including the 80% spread of this value for individual members. The black bar in the centre shows the relative size of this linear trend to the ensemble spread. (b,d) Global map of the ratio between the ensemble linear trend in near-surface temperature and the ensemble standard deviation in near-surface temperature. (a,b) show the PD time slice, (c,d) the 2K time slice.

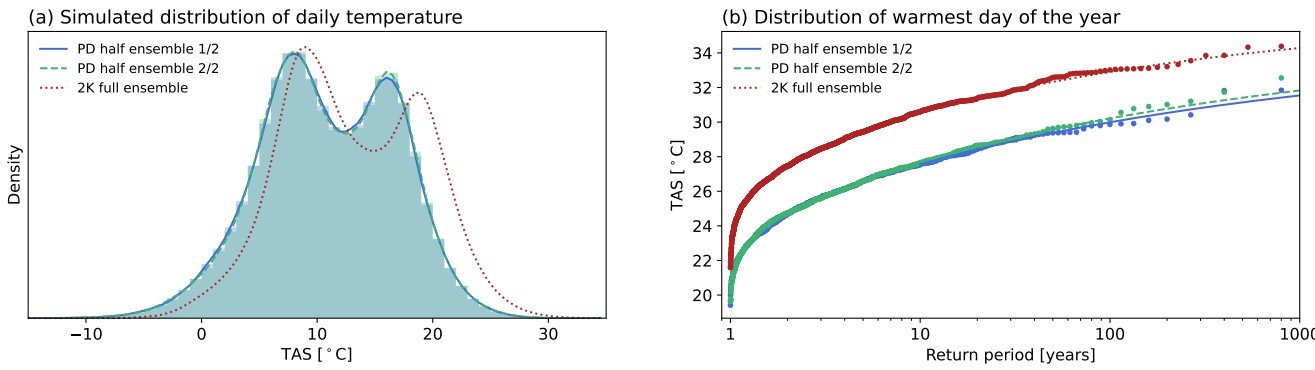

**Figure 4. Quantification of assumption 2: sampling internal variability within a time slice.** (a) Distribution of daily 2 m temperature (TAS) data at 52.3°N, 4.9°E, in two halves of the PD ensemble (blue and green shading and lines, each based on 800 years) and 2K ensemble (red dotted line, based on 1600 years). (b) GEV fit distribution (lines) and modelled data (dots) value plot for the warmest day of the year, using the same colours as in (a).



## 3.5 Legacy of micro perturbations in simulated variability

Ensemble members from a common parent have large similarities in the first days/months/year of the simulation. The initial conditions of ocean are completely the same. Only in the atmosphere there are small differences due to the micro perturbation.
The effect of the shared parent on variability differs per location and variable. We test this with a subset of parents for local near-surface temperature (TAS) variability in De Bilt, The Netherlands, and for El Niño–Southern Oscillation (ENSO) variability. The TAS variability seems to have lost the initial-condition information after around 20 simulation days (Figure 5). For the ENSO signal it takes 2 or 3 years to lose the initial-condition information (Figure 6).

The legacy of information from the initial conditions has consequences for the estimated variability. This may show a
265 spurious peak at this (early) year 1 value. We therefore advise future users to quantify this effect for their variables of interest, and if necessary remove the first days/months/year of each simulation to ensure that estimates of variability do not suffer from this effect.

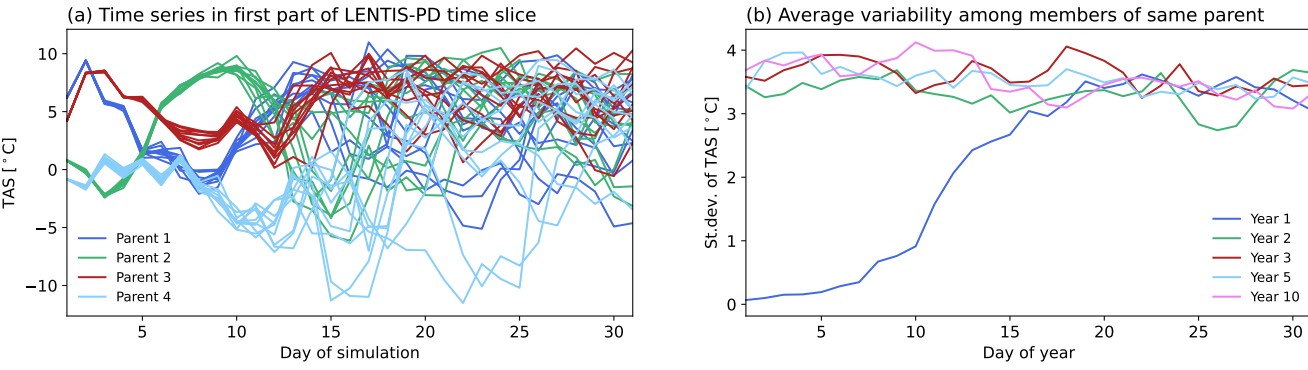

**Figure 5. Test of influence of parent on variability: local TAS.** (a) Time series, coloured by parent, of TAS at a grid point (52.3°N, 4.1°E) for the first 31 days of the simulations. (b) Time series of variability (estimated as the standard deviation of TAS of all members with a certain parent, then averaged over the parents) different years in the time slice.

## 4 Demonstration

In this section, we demonstrate the ensemble by giving examples of interesting cases. This ensemble has the unique feature of
270 high frequency output, allowing for detailed studies into extreme events. The surface water balance and surface energy balance variables are stored at 3-hourly intervals. Atmospheric variables (relative humidity, specific humidity, temperature, eastward wind, northward wind, omega) are saved daily on eight pressure levels (1000, 850, 700, 500, 250, 100, 50, 10 hPa). Additionally, a number of land, ocean, atmosphere variables is stored monthly. The variables are post processed and standardized to CMIP6 convention. The full overview of the output variables can be found in Appendix B. More information on the variables and




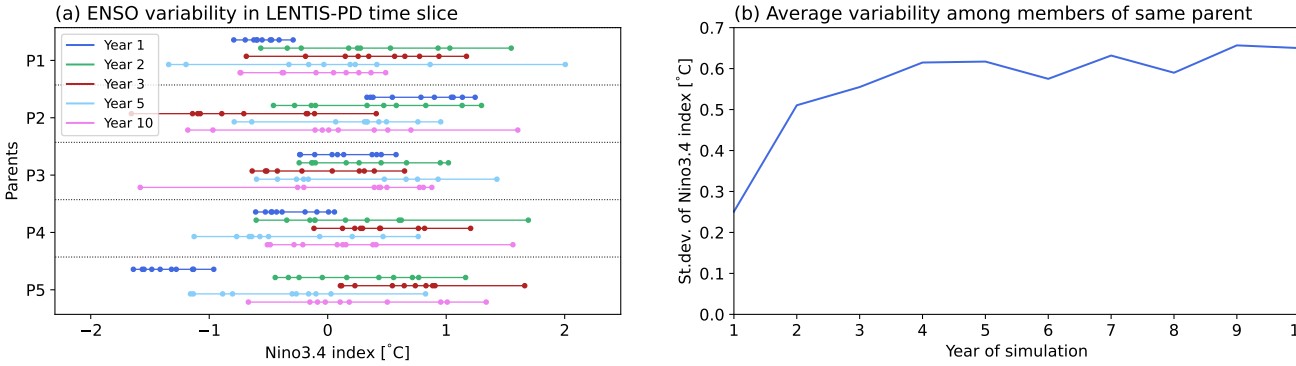

**Figure 6. Test of influence of parent on variability: Niño-3.4.** (a) Distribution of the annual mean Niño-3.4 index, separated by parent and simulation year. (b) Time series of variability (estimated as the standard deviation of Niño-3.4 of all members with a certain parent, then averaged over the parents) throughout the time slice.

their output dimensions is accessible via the following search tool: https://clipc-services.ceda.ac.uk/dreq/mipVars.html, last accessed September 19, 2022.

## 4.1 Climatological context of observed extreme events

Weather extremes usualy take place in a point in time. This often raises questions about the climatological context of such an event and about the climate change aspects of similar events. The KNMI-LENTIS data set can be used to determine, for
example, the return interval of specific type of events. With this information we can infer the range of frequency and magnitude that can occur within the climate's variability, and what the influence of climate change is on this (van Oldenborgh et al., 2021; van der Wiel et al., 2021).

The effect of the forced climate difference can be seen in other variables throughout the Earth system since EC-Earth is a fully coupled climate model. Figures 7c-e highlight some extreme weather/climate events that have occurred in recent past. The
Greenland Ice Sheet (GrIS) has seen unprecedented melt events in the recent years (e.g. [http://nsidc.org/greenland-today/2021/08/large-melt-event-changes-the-story-of-2021/], last accessed: Oct 3, 2022). Figure 7c shows the simulated return intervals of July average snow melt rates for a grid point in the eastern part of the Greenland Ice Sheet (72°N, 30°W). The higher return intervals in the 2K time slice are due to a projected increase in future July melt event frequency.

The Southern England railroad organisation introduced speed restrictions during the 2022 heat wave in response to drying
and shrinking of the clay soils, which made the train tracks prone to movement (e.g. [https://www.networkrail.co.uk/stories/soil-moisture-deficit-on-the-railway/], last accessed: Oct 3, 2022). Figure 7d shows simulated surface air temperatures in July for a grid point in Southern England (51°N, 2°W) against column integrated soil moisture content, for the PD and 2K time slices. The upward-leftward shift of the scatter cloud indicates more co-occurrences of hot temperature - soil moisture deficit events in the warmer climate.





Finally, the World Meteorological Institute foresees "a strong chance of drier than average conditions in most parts of the Horn of Africa, making this the fifth consecutive failed rainy season" for the October-December 2022 season (e.g. [https://www.africanews.com/2022/08/26/horn-of-africa-5th-consecutive-rainy-season-missed/], last accessed: Oct 3, 2022). Figure 7e shows the simulated seasonal cycle of precipitation in a grid point in the Horn of Africa (8°N, 48°E). In contrast to the news item, the simulated warmer climate appears to become generally wetter, with shifts in the start and intensity of the

rain season. However, further analysis is required to draw conclusions on the occurrence of droughts.

## 4.2   Added value for extreme meteorological event research

Grey and blacks swans are terms for the types of extreme events that are yet unobserved or cannot even have been anticipated, respectively (Watkins, 2013; Aven and Renn, 2015). These type of events by definition have no evidence in historical observations. Robust statistical analyses are often not possible due to their sparsity, not even with large ensemble data. A recently

developed technique into swan-type events is using story lines (Shepherd et al., 2018; Lloyd and Shepherd, 2021; Sillmann et al., 2021). By composing a story line the extreme event can be understood in terms of its spatial and temporal meteorological context. This can also help to gain understanding of such events in a different climate. KNMI-LENTIS provides a physically coherent data set that allows for story-line type of research into possible extreme meteorological events in the present day and in a 2K warmer climate (van der Wiel et al., 2021).

In this example we identify the hottest day in De Bilt (52.3°N, 4.9°E), the Netherlands in the PD time slice and its meteorological circumstances. The daily maximum near-surface air temperature reaches 39°C on this day. The hottest day to date in the observed records at the KNMI was measured on 2019-07-25, with a peak temperature of 37.5 °C (https://www.knmi.nl/over-het-knmi/nieuws/warmste-dag-van-het-jaar-nu-4-c-warmer-dan-rond-1900, last accessed: Oct 4, 2022), demonstrating that the ensemble can indeed be used to study events beyond the observed record. Figure 8a shows very high

maximum temperatures in a large area of west and central Europe and across the Mediterranean. The physical drivers of such an extreme event can be both large scale warm air advection and local land-atmosphere processes. The evolution of surface energy balance components in De Bilt (Figure 8b) do not suggest that local land-atmosphere processes are a main contributor to this heat event. The sea level pressure field over the Northern Hemisphere points to advection of hot air from the south (Figure 8c). A small low-pressure system west of the British Isles seems important in directing the flow northward. We note

that temperatures in northern America are anomalously high as well (not shown). Further analysis is needed to assess whether the extremely hot weather is related to the specific configuration of high and low pressure systems that is seen in earlier studies in connection with simultaneously occurring heat waves in the Northern Hemisphere (Kornhuber et al., 2019).

## 4.3   Added value for compound event research

The multivariate nature of compound events, events where combinations of climate drivers and/or hazards contribute to societal

or environmental risk (Zscheischler et al., 2018), requires a bottom-up approach in which all data are physically consistent. Output from climate models is by definition physically consistent, though when bias corrections or statistical extrapolations are applied this consistency may be broken (i.e., broken consistency between variables or lost consistency in time due to e.g.





**Figure 7. Ensemble climatological context.** For PD in blue and 2K in red: (a) histogram of annual mean Global Mean Surface Temperature (GMST), (c) return interval in years of surface snow melt rates in East Greenland Ice Sheet grid point (72°N, -30°E), (d) scatter plot of total column integrated soil moisture content and near-surface temperature in Southern England grid point (51°N, -2°E) with the cloud mean as dot and 2 standard deviations in the ellipse, (e) annual cycle of precipitation in Horn of Africa grid point (8°N, 48°E), with box plots for the ensemble spread defined as box: first Quartile - Median - third Quartile ( Interquartile Range (IQR)), whiskers: first Quartile - 1.5×IQR and third Quartile + 1.5×IQR and outliers: values outside of these limits. EC-Earth3 fixed land surface orography and ocean bathymetry (m) in (b) is the same for both PD and 2K.




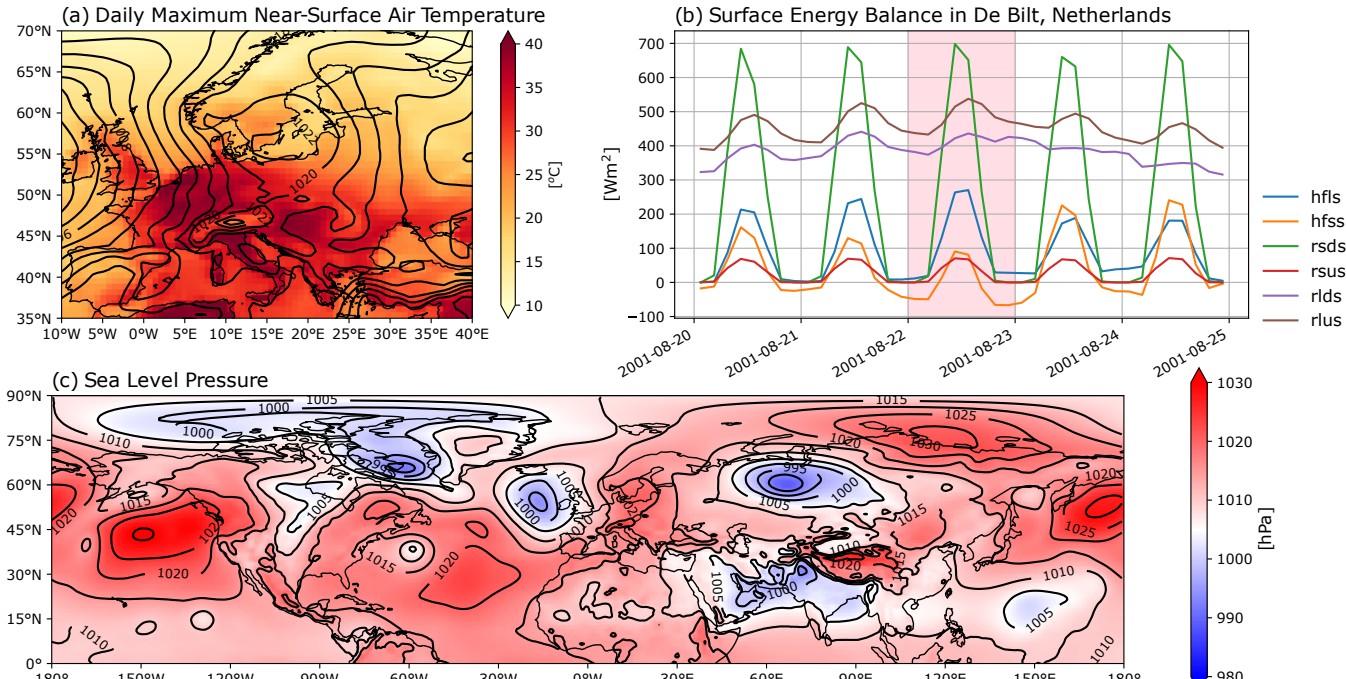

**Figure 8. Hottest day in De Bilt, The Netherlands, in the PD ensemble.** (a) Europe Maximum surface air temperature over Europe on the hottest day (colors) and sea level pressure (contours every 5 hPa). (b) Surface energy balance components in De Bilt, The Netherlands, during the days around the hottest day (pink shading) with the latent heat flux (hfls), sensible heat flux (hfss), downwelling shortwave radiation (rsds), upwelling shortwave radiation (rsus), downwelling longwave radiation (rlds) and upwelling longwave radiation (rlus). (c) Northern Hemisphere Sea level pressure in colors and in contours every 5 hPa.

a no longer closed water budget). Time slice large ensembles are very suitable for the analysis of rare or extreme compound events (e.g., Kelder et al., 2022), owing to the physical consistency of the data, and the explicitly resolved extreme events due

to the size of the ensemble. In this section we demonstrate this using a case study on the extreme wheat yield loss in France.

The 2016 winter wheat harvest in France was exceptionally low (28% lower than the expected value) and Ben-Ari et al. (2018) showed that this was caused by the "compound interaction between temperature in the late autumn/early winter and precipitation in the spring". 2016 was unique in the combination of low exposure to cold days in autumn ('vernalizing days', days with maximum temperatures between 0 and 10°C) followed by wet spring conditions (high precipitation). The historic

record, here shown through ERA5 reanalysis data (Hersbach et al., 2020; Bell et al., 2021), shows how exceptional the year 2016 was in terms of these variables and especially in their multi-variate combination (Figure 9).

The KNMI-LENTIS PD ensemble provides many more samples of winter-wheat growing conditions (1440 simulated seasons) than the observed historical record. The simulated data includes some seasons with similar or even more extreme compounding conditions than the 2016 observed event (Figure 9). This provides an opportunity to better understand the relationship

between the governing variables, and investigate (remote) drivers of compounding conditions. Note that biases in the simulated




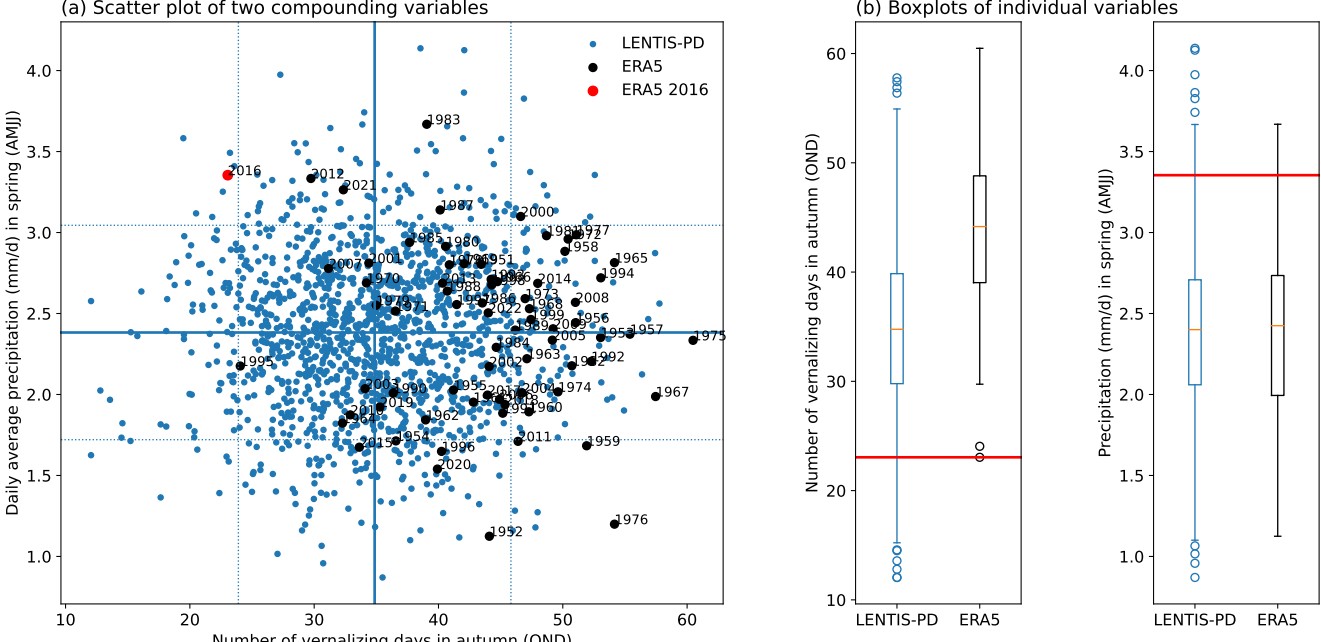

**Figure 9. Example compound event research.** Analysis of meteorological circumstances that lead to extreme wheat yield loss in France after (Ben-Ari et al., 2018). Presented values are the area average of the Northern part of France (47-50°N, 0-7°E). (a) Scatter plot of number of vernalizing days in autumn (October-December) versus daily average precipitation in spring of the next year (April-July). Large black dots are ERA5 values (1950-2021), where 2016 is marked as a red dot. Small blue dots are KNMI-LENTIS-PD values, horizontal and vertical lines correspond to the average (solid lines) ± 1 standard deviation (dotted lines). (b) Box plots of the number of vernalizing days and the daily average precipitation for both KNMI-LENTIS PD and ERA5.

data (here we find the model has a low bias in the number of vernalizing days) can impact estimates of event likelihood and process understanding.

This case study highlights the strength time slice large ensemble data in compound event research. Also for other types of compound events (pre-conditioned events, multi-variate events, temporally compounding events and spatially compounding events; Zscheischler et al., 2020) large ensemble data can help to, for example, quantify event likelihood and identify drivers and modulators of events (Bevacqua et al., 2021).

## 4.4 Added value for climate-impact modelling

Climate science, apart from aiming to improve our scientific understanding of the physical Earth system, also aims to inform society and policy makers of (future) risks caused by adverse weather. It is during extreme events that such risks are highest. However, extreme weather events (e.g. the hottest or wettest days) do not necessarily link 1-to-1 to extreme impact events (highest heat stress or biggest floods, e.g. van der Wiel et al., 2020). This is due to the complex non-linear relationships





between weather and impacts. In this section we demonstrate this phenomenon and show that an approach of 'ensemble climate-impact modelling', as for example proposed by van der Wiel et al. (2020), enhances our understanding of the weather-impact relationship and improves estimates of (changing) societal risk.

For this case study, we simulate electricity production from solar radiation (Photovoltaics, PV) with a relatively simple model that considers incoming solar radiation, near-surface temperatures and near-surface wind speeds (Jerez et al., 2015; van der Wiel et al., 2019b). We use daily data at a single grid point in the Netherlands (De Bilt, 52.3°N, 4.9°E). Higher values of incoming radiation in summer (sunnier conditions and relatively long daylight hours) lead to higher values of PV generation, but the heating of solar cells negatively impacts generation (heating mostly related to high temperatures, some cooling provided

by wind). We computed PV potential for all days in the KNMI-LENTIS PD ensemble (1600 years, > 0.5 million days). Here, we investigate the relationship between meteorological variables and PV potential, and the timing of extreme production days.

     As expected, PV potential is strongly related to incoming solar radiation (Figure 10a). The histogram shows a cluster for both DJF and JJA, indicating that solar cells work more efficiently in winter. This is due to differences in solar cell heating and daylight hours: for 100 W/m$^2$ incoming solar radiation, the PV potential in DJF is approximately 27% whereas in JJA it

is approximately 18%. On the other hand, the large seasonal difference in incoming radiation at this latitude makes summers about 5 times more productive in terms of PV (Figure 10b). Therefore, extreme production events, i.e. days with extremely high PV potential, are expected to occur in late sping – early summer. Indeed the annual maximums of PV potential occur in early JJA, and they do not coincide with the annual maximum of incoming solar radiation (Figure 10b) which occur later.

     This case study highlights some of the possibilities of ensemble climate-impact modelling. Though extreme PV production

370 days are not likely to put society at risk, the (temporal) disconnect between weather extremes on the one hand and impact extremes on the other hand is obvious. As shown in earlier sections, large-ensemble climate modelling can considerably contribute towards understanding events in the tail of the distribution. This is true for meteorological extremes (e.g. Section 4.2) but equally so for climate-impact extremes that are more closely related to possible natural or societal impacts/risk.

## 5   Conclusions

We have presented the KNMI Large ENsemble TIme Slice (KNMI-LENTIS): a new large ensemble data set produced with the re-turned version of the global climate model EC-Earth3. The time-slice approach is different from the more traditional transient ensembles available from other institutions. The advantage is that the signals of natural variability and climate change are not mixed, due to our assumption that the forced change in a slice is small. Therefore, the variability we see in a time slice is only natural variability at a given GMST level, and does not include a climate change signal. This renders our data

set specifically suitable to study climate variability and changes therein between the present-day climate and a warmer future climate. Furthermore, the data set is particularly geared towards research into land-atmosphere interface, with 3-hourly output of the surface water balance and surface energy balance variables.

     The ensemble consists of two distinct time slices: a present-day time slice and a +2K warmer future time slice relative to present-day. The present-day time slice is represented by the years 2000-2009 and forced with CMIP6 historical forcing. The





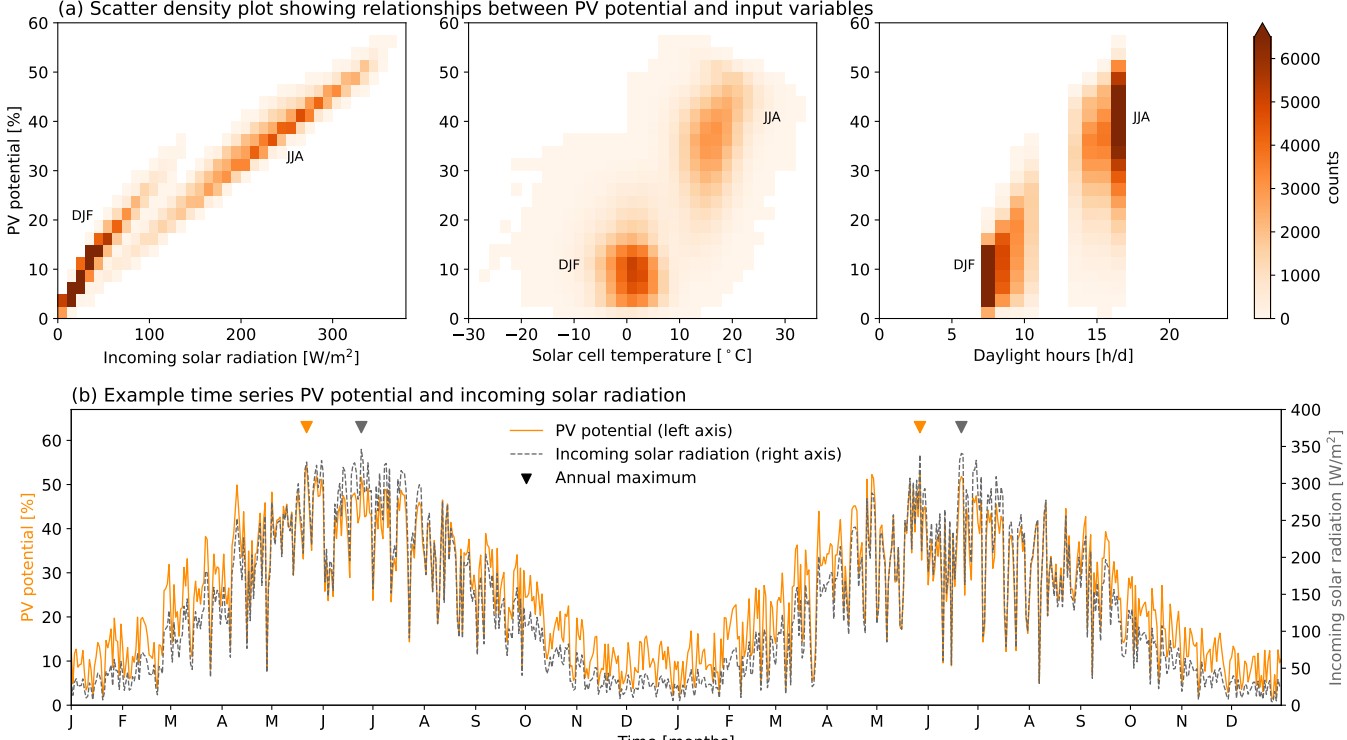

**Figure 10. Example ensemble climate-impact modelling.** (a) Scatter-density plots showing the relationships between PV potential and other variables in DJF and JJA. From left to right: incoming solar radiation, solar cell temperature, and daylight hours. (b) Example time series of PV potential and incoming solar radiation. Triangles at the top show timing of annual maxima.

+2K time slice is represented by the years 2075-2084 and forced with CMIP6 SSP2-4.5 forcing. Each time slice consists of 1600 simulated model years in 160 segments of 10 years.

The initial conditions for the ensemble members are generated with a combination of micro and macro perturbations. We have quantified the assumptions underlying the set-up, which are that the time slice simulation length is small enough so that a forced climate change signal is minor is most cases, and that the ensemble size is sufficient to sample the full distribution of

390 climate variability. We have provided examples of how this ensemble can be used to demonstrate its added value for extreme and compound event research and for climate-impact modelling. The model and thus our data set has a considerable warm bias in the Southern Ocean and over Antarctica. Future users of KNMI-LENTIS are advised to make in-depth comparisons with observational or reanalysis data especially when their studies focus on ocean processes, on locations in the Southern Hemisphere or on teleconnections involving both hemispheres.



*Code availability.* The KNMI-LENTIS production scripts are recorded on Zenodo (Muntjewerf et al., 2023b).The EC-Earth model is restricted to institutes that have signed a memorandum of understanding or letter of intent with the EC-Earth consortium and a software license agreement with the European Center for Medium-Range Weather Forecasts (ECMWF). Confidential access to the code and to the data used to produce the simulations described in this paper can be granted for editors and reviewers.

*Data availability.* The KNMI-LENTIS dataset description is recorded on Zenodo (Muntjewerf et al., 2023a), providing details of the layout of dataset, where it is located, how it is stored and how one gains access. At a later stage, (part of) the data may be made publicly available from the Earth System Grid Federation (ESGF) data portal.

*Author contributions.* LM performed the simulations; LM and TR designed the framework for the data production and validation; LM and KW performed the analyses and prepared the initial draft of the manuscript; KW, LM, RB designed the ensemble protocol; KW and RB conceptualized the study and acquired funding. All authors contributed to the final paper.

*Competing interests.* The authors declare there are no competing interests.

*Acknowledgements.* The production of the KNMI-LENTIS ensemble was funded by the KNMI multi-year strategic research funding MSO-VAREX.



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



**Table A1. Naming convention of LENTIS members.** The simulations are named with a 4 digit name: *kllm*. 'k' is a placeholder to denote the start year. Options are 'h' for 2000-historical and 's' for 2075-SSP2-4.5. 'll' is a placeholder to denote the parent. Parents run from 01 to 16, from which full transient simulation the initial conditions are taken. 'm' is a placeholder to denote the seed. Seeds run from 0 to 9, corresponding with the randomising seed of the micro-perturbation.

| ll (parent) (→) <br> m (seed) (↓) | k01m | k02m | ... | k16m |
|---|---|---|---|---|
| kll0 | k010 | k020 | ... | k160 |
| kll1 | k011 | k021 | ... | k161 |
| ... | ... | ... | ... | ... |
| kll9 | k019 | k029 | ... | k169 |

**Appendix A:  Ensemble label and CMIP6 variant label of KNMI-LENTIS simulations**

All ensemble simulations have a unique name that reflects the start year and forcing, the parent and the seed (Table A1). The
start year is noted in the first digit of the KNMI-LENTIS simulation name: 'h' for the PD time slice and 's' for the 2K time slice. The parent is reflected in the second and third digit. The seed is reflected in the fourth digit.

Further, all KNMI-LENTIS simulations are labeled per the CMIP6 convention of variant labelling. A variant label is made from four components: the realization index $r$, the initialization index $i$, the physics index $p$ and the forcing index $f$. Further details on CMIP6 variant labelling be found in *The CMIP6 Participation Guidance for Modelers*: https://pcmdi.llnl.gov/CMIP6/
Guide/modelers.html, last accessed September 20, 2022).

In the KNMI-LENTIS data set, the forcing is reflected in the first digit of the realization index $r$ of the variant label. For the historical simulations, the one thousands (r1000-r1999) have been reserved. For the SSP2-4.5 the five thousands (r5000-r5999) have been reserved. The parent is reflected in the second and third digit of the realization index $r$ of the variant label (r?01?-r?16?). The seed is reflected in the fourth digit of the realization index $r$: (r???0-r???9), The seed is also reflected in
the initialization index $i$ of the variant label (i0-i9), so this is double information. The physics index p5 has been reserved for the ECE3p5 version, so all KNMI-LENTIS simulations have the p5 label. The forcing index $f$ of the variant label is kept at 1 for all KNMI-LENTIS simulations. As an example, variant label r5119i9p5f1 refers to: the 2K time slice with parent 11 and randomizing seed number 9. The physics index is 5, meaning the run is done with the ECE3p5 version of EC-Earth3.



## Appendix B: List of output fields

The list below contains all output variables of KNMI-LENTIS data set, sorted per output frequency. See the Supplementary

file **Appendix_request-overview-CMIP-historical-including-EC-EARTH-AOGCM-preferences.pdf**