# Peer review of "The KNMI Large Ensemble Time Slice (KNMI-LENTIS)"

_EGUsphere, 2022_

## Author Comment (AC1)

**RC1**

This paper introduces the new time slice large ensemble from KNMI. It is well written, describes the experiments in details and presents some interesting applications. I recommend it is accepted with minor revision as detailed below.

**We thank the reviewer for their kind words and critical review. Below, we respond to their remarks in blue.**

45 change e.g. to 'for example'
**Changed**

Reference issue line 64
**Solved**

Line 60 Could reference introduction paper to the large ensemble special issue: https://esd.copernicus.org/articles/12/401/2021/ and/or single forcing large ensembles e.g.: https://www.cesm.ucar.edu/working-groups/climate/simulations/cesm1-single-forcing-le and https://open.canada.ca/data/en/dataset/aa7b6823-fd1e-49ff-a6fb-68076a4a477c
**Added**

Line 71 – this seems like a good assumption but can you test it? Perhaps look at 1300 vs 1600 and see how different they are. After reading the paper I see the authors do test this. Perhaps allude to it here on line 71
**We added a line of text for clarification and a reference to the section that tests the assumptions.**

76 be consistent – should be 5 years
**Changed**

149 – have you checked that there is no trend?
**Yes, we demonstrate the trend in GMST in the two time slices in figure 3a and c, which are 0.20 K/10yr and 0.22 K/10yr respectively.**

174 – could replace 'means' with 'refers to'
**Changed**

178 – could replace 'like' with 'such as'
**Changed**

3 E can you suggest the ensemble mean is removed for variables where trend is too large as in many other le studies
**We appreciate the reviewers' suggestion. Our set-up aims to facilitate both researchers interested in multi-years phenomena, which will have a small present-day and +2K climate change trend, as well as researchers interested in sub-annual events. Of course, users can**

**detrend if that fits their research needs. But for us to advise to do so would be disregarding of the former group.**

Section 3.5 you could cite and discuss the following papers:
https://wires.onlinelibrary.wiley.com/doi/full/10.1002/wcc.563
https://link.springer.com/article/10.1007/s00382-015-2806-8
https://journals.ametsoc.org/view/journals/clim/36/2/JCLI-D-21-0176.1.xml
**We thank the reviewer for suggesting to discuss this topic. We added a section on irreducible uncertainty at the end of this section.**

262 can you make a stronger statement than 'seems to'
**We are adding clarifying sentences. With the examples that are provided, we aim to illustrate how quickly the chaotic nature of the Earth system model takes over the initial condition micro-perturbation. We don't intend to make an absolute statement here, as the speed of dispersion can vary spatially. Therefore, we advise users to quantify this effect specifically for their research purposes.**

589 ??? Are confusing
**Yes, I can understand that. We have replaced '?' with 'X' in the hope that this is a more intuitive way to convey the concept.**

4.1 I don't see a reference to fig a/b
**For figure 4 indeed misses the reference to panel b. This is added.**

Fig 7 no reference to b after a
**Yes, that is correct. We have removed figures 7a+b.**

Figure 10 define PV in the caption please
**Changed**

[revised manuscript text omitted]
 advantage of the time slice approach is that is has much larger sample of climate variability: with 160 members of each climate, only the Max Planck Institute Grand Ensemble (MPI-GE, Maher et al. (2019)) comes close in magnitude with its 100 members. The disadvantage of the time slice approach is that we are limited to comparing two more or less static different climates, while the transient approach allows to assess a large temporal range of climate change. Time slice large ensembles are not yet common in contempory climate change modelling; we know of two earlier time slice ensembles: one with EC-Earth2~~

[revised manuscript text omitted]

---

## Author Comment (AC2)

**RC2**

Review of "The KNMI Large Ensemble Time Slice (KNMI-LENTIS)" by Muntjewerf et al.

**General comment:**

The manuscript presents the KNMI Large Ensemble Time Slice, which consists of a present-day 2000-2009 and a future +2K 2075-2084 time slice with 160 ensemble members each generated by micro and macro perturbation. The manuscript is well structured and clear, with high scientific rigour, and represents an important contribution not only to the large ensemble community. I much appreciate that the authors test the assumptions they do and that limitations are discussed explicitly. The manuscript further demonstrates the added value of KNMI-LENTIS for extreme and compound event research and for climate-impact modelling. I find these application examples appealing and congratulate the authors on this manuscript. However, I do have a few comments that need to be addressed before I can recommend publication. These comments only require revision of the text.

Recommendation: Minor revisions

**We thank the reviewer for their kind words and critical review. Below, we respond to their remarks in blue.**

**Specific comments:**

l. 3: What is a re-turned version? I assume this should be "re-tuned". Same typo in l. 376. **Changed**

l. 13: The authors should add here that the tuning is done to reduce the Eurasian warming bias, and the focus region tuned for is Europe/Eurasia. This aspect should be mentioned in the Abstract and generally highlighted more in the text because it limits the usability for studying climate extremes for the large part of the globe that shows substantial model bias even on low and mid-latitude land regions (see Figure 2a).
**We thank the reviewer for this point. The retuning was targeting the Northern Hemisphere, not specifically Europe, but we can see how that idea can come about. We have been very Europe-centric in the analysis and the examples in the demonstration. In the manuscript we add bias quantification of N-America and S&SE Asia regions in figure 2, table 1 and table 2. For these regions, like for Europe, the GMST bias w.r.t. ERA is quite good (<0.5K)**

ll. 54-60 and ll. 376-381: The second lines mentioned here are a good attempt to compare the time slice approach with the large ensemble approach of transient simulations. However, the introduction misses a more elaborated comparison between the advantages and disadvantages of the two approaches. For instance, the advantage mentioned here: "This renders our data set specifically suitable to study climate variability and changes

therein between the present-day climate and a warmer future climate." is even more true for a transient ensemble. I suggest to use the paragraph in ll. 54-60 to provide more details on 1) disadvantages of the time slice approach compared to the transient approach, 2) why the authors decided to use the time slice approach (I assume computational or data storage restrictions but this is not mentioned explicitly), 3) whether there are other time slice large ensembles than from KNMI, and 4) that the approach of time slices is extensively used in paleo-climate modelling. Please revise and expand the introduction for a more complete background on the time slice approach.

**1+2. Adding to paragraph in II. 54-60:**
**The advantage of the time slice approach is that is has much larger sample of climate variability: with 160 members of each climate, only the Max Planck Institute Grand Ensemble (MPI-GE) comes close in magnitude with its 100 members. The disadvantage of the time slice approach is that we are limited to comparing two more or less static different climates, while the transient approach allows to assess a large temporal range of climate change.**

**We have elaborated more on irreducible uncertainty, forced climate change and climate variability in the introduction, in section 3.5, and in the conclusion.**

**We disagree with the statement of the reviewer**: *"This renders our data set specifically suitable to study climate variability and changes therein between the present-day climate and a warmer future climate." is even more true for a transient ensemble.* **Their underlying assumption is incorrect: we did not develop this design for reasons of computational or data storage limitations. KNMI-LENTIS encompasses a large amount of data (more than 120 TB). In comparing the time slice ensemble to a transient ensemble in their use to study climate variability and changes therein, we need to consider only consecutive years with equally small trend and climate change signal. If we consider the entire transient run and look at how the variability changes, we have a mixed signal: the forced change in mean climate, the forced change in internal variability and the state of climate at that moment. With the time slice ensemble the forced change in mean climate can more easily be removed, and to separate the latter two it has many more members (160) than any transient large ensemble and therefore is more suitable for these types of studies.**

**3. We have added 1 reference to a time slice ensemble we know, that was done with HadGCM2 in a project together with the predecessor of KNMI-LENTIS.**

**4. With regards to paleo modelling: as far as we understand time slices in this field are used as means to compare different glacial or interglacial periods. Paleo large ensembles are generated usually with climate models of intermediate complexity, and by perturbing model parameters rather than initial conditions. While we do see similarities with the time slice use in the paleoclimate community and appreciate the suggestion to include this topic, we see also a lot of differences and we choose not to include these remarks in the manuscript to avoid confusion.**

l. 67 and l. 85: I recommend not to use the term data set ifor model simulations because data sets rather refer to observational data. I suggest to change "transient ensemble data

sets" to "transient ensemble simulations" and "data set" to "ensemble" here and in other places.
**Changed**

ll. 83-84: It is not clear to me why the decade 2000-2009 is used for the "present-day" time slice, 2010-2019 would have better allowed for comparisons of present-day vs 2075-2084. I understand the two reasons the authors mention (historical CMIP6 forcing ending in 2014, initial condition files only every 10 years), but then the term "present-day" is misleading. I suggest to call it "past decade" or similar instead.
**We appreciate the reviewer pointing out this subject and we can understand where they're coming from in suggesting to change the name of the period. However, we do not agree the term 'present-day' is misleading. 2000-2009 lies in the middle of the latest WMO-defined climate normal 1991-2020, and therefore this decade can reasonably be considered present-day. We add this argument in the manuscript.**

**Also, we have recalculated the biases for the period 1991-2020 (instead of 1990-2019), for consistency.**

ll. 144-146: The manuscript misses to mention how KNMI-LENTIS can be compared to the SMHI Large Ensemble (SMHI-LENS) with EC-Earth3.3.1 by Wyser et al., 2021. I think I understood that KNMI-LENTIS is not directly comparable to that model version used for the 50-member ensemble because of the performed re-tuning but can be directly compared to the CMIP6 16-member ensemble. The authors mention that "The ECE3p5 version is a re-tuned version of the EC-Earth 3.3. release for CMIP6 (Döscher et al., 2021)". However, the authors do not mention how exactly the re-tuning affects/prevents the comparison to SMHI-LENS, whether there are other differences than the re-tuning, e.g. from a different model version? I like to encourage the authors to elaborate on the comparability of the presented time slices with the 50-member SMHI-LENS.

**The difference between SMHI-LENS and KNMI-LENTIS is that these ensembles have a different equilibrium climate: that affects the climate in the pre-industrial, historical and SSP simulations.**
**We include in the text calculations of ECS and TCS of EC-Earth3_p1 (from SMHI) and EC-Earth3_p5 (KNMI). The transient climate response (TCR) is 2.1 voor ECE p5 (2.3 voor ECE p1 en 2.0 voor CMIP6 multimodel mean. The model's effective climate sensitivity (ECS) is 4.0 voor ECE p5 (4.3 voor ECE p1 en 3.7 voor CMIP6 multimodel mean.)**

**The model version of SMHI-LENS is the CMIP6 release version of EC-Earth3: it is tuned to have the smallest possible GMST bias with respect to reanalyses and observational records (a.o.). Our version on the other hand is tuned to have the smallest possible Northern Hemisphere-MST bias.**
**Other than that, there are very few differences between the models: the model version used for KNMI-LENTIS (e.g., the atmosphere & ocean dynamical core, the land- and sea-ice models) is the same as EC-Earth3.3.1 that is used for SMHI-LENS.**

**For the overlapping years and scenario forcing, KNMI-LENTIS can be compared to SMHI-LENS like any other LENSes with common model ancestry.**

**We add text to the manuscript to describe this point.**

ll. 159-161: I am confused here. I don't understand why the period 1985-2014 is taken to calculate the present-day mean state, not the first time slice 2000-2009 as done otherwise? Please clarify.
**Answering together with the comment below.**

l. 169: The authors term the period 2075-2084 as +2K in the SSP2-4.5 scenario but state here that +2K are reached in year 2073. While keeping the term +2K, please specify the exact average warming of the 10 year period compared to present-day. I am further confused by the value of annual GMST increase of 1.95K in Table 1 and in l. 208. According to the text, I understand the exact value should be above +2K. Please revise this section on the apparent difference between 1985-2014 and 2000-2009 for defining +2K.
**We can understand the reviewers' confusion here. The chosen periods and the 2K SSP-forcing are the outcome of a set of choices that evolved over the process of designing the LE. We are have revised the text to reflect this decision-making process better.**

**For the estimate of the PD climate, we took a 30-year climatology as is a common length of time to define a climate state. We chose the year 1985-2014 from the 16 ECE3_bis historical members. This period is exclusively forced by historical forcing, so we avoid blending in a SSP scenario after 2014 and nudging our analysis in a particular direction. In the analysis there were 2 things that had to align:**
**1) which SSP scenario the use while keeping the decadal climate change signal similar to the PD signal, and**
**2) in which year to start.**
**Our main criterium to make a decision was the decadal climate change trend. Further limitations are technical: the availability of initial conditions, and the need to have one forcing scenario per time slice.**

**Based on this, we found the best option was to use SSP2-4.5 scenario. It reaches 2K on average year 2073 (with a std.dev of 9.5 years) in the 16 ECE3_bis members.**
**Because the spread in timing is so large (because the forced signal is relatively small), this allowed us some flexibility to find two periods within our technical constraints.**
**So, we've made a best estimate of what time periods, given these technical constraints, would yield a +2K difference: shifting both PD and 2K forward a few years in a way that matched our available input files.**

**In table 1 we assess what deltaGMST we actually ended up with in LENTIS, after running all the simulations. It turns out that it's close to our target of 2K, namely 1.95. We are very happy with that.**

Figure 7 panel b is only described at the end of the caption and it is not clear to me why the orography and bathymetry matters in the context of that application. I would suggest to remove it from Figure 7.

**Yes, that is correct. We have removed figures 7a+b.**

Technical corrections:

l. 4: ten years each
**Changed**

l. 8: specify sub-annual timescales
**Changed to sub-daily**

l. 47: 2x the
**Changed**
l. 70: simulation years
**Changed**

l. 153: 2x annual
**Changed**

l. 343: This is not a proper sentence.
**Changed**

[revised manuscript text omitted]
 advantage of the time slice approach is that is has much larger sample of climate variability: with 160 members of each climate, only the Max Planck Institute Grand Ensemble (MPI-GE, Maher et al. (2019)) comes close in magnitude with its 100 members. The disadvantage of the time slice approach is that we are limited to comparing two more or less static different climates, while the transient approach allows to assess a large temporal range of climate change. Time slice large ensembles are not yet common in contempory climate change modelling; we know of two earlier time slice ensembles: one with EC-Earth2~~

[revised manuscript text omitted]